# Visual Detection of *Clostridium perfringens* Alpha Toxin by Combining Nanometer Microspheres with Smart Phones

**DOI:** 10.3390/microorganisms8121865

**Published:** 2020-11-26

**Authors:** Aiping Cao, Heng Chi, Jingxuan Shi, Ruiqi Sun, Kang Du, Yinna Song, Min Zhu, Lilin Zhang, Jinhai Huang

**Affiliations:** 1School of Life Sciences, Tianjin University, Tianjin 300072, China; happycap@tju.edu.cn (A.C.); chiheng@tju.edu.cn (H.C.); shijingxuan@tju.edu.cn (J.S.); srq_77@tju.edu.cn (R.S.); 2018226023@tju.edu.cn (Y.S.); 15195809381@163.com (M.Z.); 2School of Precision Instruments & Opto-Electronics Engineering, Tianjin University, Tianjin 300072, China; dukang@tjbskj.cn

**Keywords:** *C. perfringens*, alpha toxin (CPA), smartphone image, nano-silica microspheres

## Abstract

*Clostridium perfringens* α toxin (CPA) is an important virulence factor that causes livestock hemorrhagic enteritis and food poisoning by contaminated meat products. In this study, the nano-silica microspheres combined with smartphone image processing technology was developed to realize real-time CPA detection. First, the N-terminal and C-terminal domain of the CPA toxin (CPA_C3_ and CPA_N_) and their anti-sera were prepared. The silica microspheres coupled with the antibody of CPA_C3_ was prepared to capture the toxin that existed in the detection sample and the fluorescent-labeled antibody of CPA_N_ was incubated. Moreover, the fluorescent pictures of gray value were performed in a cell phone app, corresponding to toxin concentration. The new assay takes 90 min to perform and can detect CPA as little as 32.8 ng/mL. Our results showed a sensitive, stable, and convenient CPA detection system, which provides a novel detection method of native CPA in foods.

## 1. Introduction

*Clostridium perfringens* (*C. perfringens*, also named *Clostridium welchii*) can result in food-borne gastroenteritis and other infectious diseases. As a conditional pathogen, *C. perfringens* causes a wide range of diseases in a variety of hosts, due to the production of a diverse set of toxins and extracellular enzymes [1]. So far, at least 20 kinds of exotoxins have been found, among which the main lethal toxins are α, β, ε, ι, *C. perfringens* enterotoxin (CPE), and novel toxin (NetB). Moreover, all types can produce α toxin, which causes hemorrhagic enteritis and acute death in livestock [2]. As the most important virulence factor of *C. perfringens* type A [3], α toxin has the characteristics of cytotoxicity, hemolytic activity, lethality, skin necrosis, myonecrosis, granulopoiesis [4], inhibition of erythroid differentiation [5], platelet aggregation, and increased vascular permeability. Besides, it can be found in the small intestines of domestic animals and can contaminate many types of retail meat products, milk, and dairy products, leading to food poisoning [6]. The activity of α toxin can be inhibited, not only by EDTA and o-phenanthroline, but also by ether-coupled phosphatidylcholine. Besides, α toxin is sensitive to pancreatin and heat, and 2.5% pancreatin can completely inactivate it at 37 °C for 1 h. When the α toxin is heated to 60–70 °C, the hemolytic activity of the toxin can be lost, and part of its activity can be restored at 100 °C [7]. According to the amino acid sequence deduced from the nucleotide sequence of the α toxin gene, the mature α toxin is 370 amino acids and consists of two domains, including the N-terminal domain amino acid (1–250 aa residues) and the C-terminal region (251–370 aa residues). The structure of α toxin has extensive homology with *Bacillus cereus* phospholipase C (PC-PLC). The PC-PLC consists of 245 amino acids and is composed of 10 β-helixes with variable α-helix connection lengths [8]. Correspondingly, it has two functional regions that the N-terminus has phospholipase C activity and the C-terminus has sphingomyelinase activity. Moreover, phospholipase C activity alone is not enough to make the α toxin toxic [9]. The mice that were immunized with the C-terminal domain of the CPA were protected against *C. perfringens* infections, and the anti-sera were able to inhibit the CPA activity [8,10]. It is an important candidate antigen for the genetically engineered subunit vaccine of *C. perfringens* type A and other clostridial toxoid vaccines [11].

The main detections of *C. perfringens* methods are immunological tests, molecular biological tests, etc. Typical identification is mainly based on the detection methods in the serum neutralization test, ELISA, and PCR. The detection of antigens is mainly aimed at the toxins secreted by various types of *C. perfringens*. In recent years, the prevalence of *C. perfringens* has become increasingly severe. With the development of molecular biology and biotechnology, various diagnostic techniques have been widely used in the diagnosis of this disease. In China, Yang et al. [12] applied SDS-PAGE electrophoresis technology to identify the type of *C. perfringens* for the first time. Hale et al. [13] proposed a capture antibody ELISA method for the determination of α toxin. This method uses standard serum as the positive antigen to be adsorbed on a solid carrier and blocks with skim milk, and then adds toxins prepared in advance. At this time, the antigen and antibody on the surface of the solid phase carrier form a complex, wash away unbound components, then add the enzyme-labeled secondary antibody, and finally add the substrate. Under the catalysis of the enzyme, the substrate will react to produce colored substances to determine the result. Lu et al. [14] used *C. perfringens* culture filtrate to obtain refined α toxin by staged salt precipitation of sulfuric acid, staged precipitation of acetone, and gel filtration. Moreover, the titers of purified α toxin and the culture filtrate of yolk were determined, respectively, by the yolk reaction turbidity (trace) method on the 96-well cell plate. The extracted *C. perfringens* type D antigens as coating antigen were used to establish an indirect ELISA method in the serological investigation of the *C. perfringens* type D infection [15]. Fach et al. first reported the use of polymerase chain reaction (PCR) to detect the α toxin of *Clostridium perfringens* [16]. Through the amplification of the α toxin gene, the gene fragments are analyzed by a restriction endonuclease, and the specificity is very high. The sensitivity for detecting bacteria in stool samples can achieve 2.95 × 10^8^ bacteria/g. When using nested PCR detection in the discussion, the detection limit for bacteria can reach 1 to 6 bacterial cells, and the results can be obtained within 2 h after amplification [17]. Moreover, some reports have developed a duplex PCR-based microsphere assay for simultaneous detection of *C. sordellii* and *C. perfringens*. The microsphere assay correctly identified *C. sordellii* and *C. perfringens* in all known isolates and in all CTS (*Clostridium sordellii* and *Clostridium perfringens* toxic shock) patients (10 *C. sordellii*, 8 *C. perfringens*, 2 both) and showed 100% concordance with PCR and sequencing results [18].

Nanomaterials have many unique properties, such as large surface area, quantum effect, a large proportion of surface atoms, high surface energy, macroscopic quantum tunneling, and other characteristics, so they have a series of optical, magnetic, thermal, biocompatibility, and catalytic properties. Nanomaterials combine biologically active molecules (such as enzymes, antibodies, DNA molecules, etc.) as signal amplification and biosensing motifs, and are used to construct molecular amplification probes for signal amplification [19]. At present, molecular recognition probes based on the signal amplification effect of nanomaterials have shown a wide range of application prospects in materials science, chemistry, medicine, bioengineering, and other fields [20,21]. Core-shell silica nanoparticles can encapsulate thousands of fluorescent dyes to obtain higher intensity and more stable fluorescent signal. Silica nanomaterials can achieve different surface functional modifications by interacting with other active molecules (such as DNA, enzymes, antibodies, peptides, carbohydrates, etc.) due to their large surface area, strong adsorption, good stability, biocompatibility, and easy surface modification. The modified DNA molecules onto silica fluorescent nanoparticles as molecular probes were used to quickly detect bacterial-related nucleic acid fragments of a variety of bacteria [22]. Usually, the antibody against protein coded by the bacterial virulence gene is modified on the surface of silica nanoparticles coating with fluorescent dyes, then the specific antibody can capture the bacteria in detected samples, the strong fluorescent signal of the silica fluorescent nanoparticles can help achieve highly sensitive detection and imaging of bacteria.

In this study, the double antibody sandwich (DAS) fluorescence microspheres method, using the nanomaterial silica microspheres coating with CPA C-domain antibody to capture native CPA in samples, the fluorescent-labeled antibody against CPA N-domain to sandwich and to display the fluorescence intensity under fluorescence microscope, were developed. The quantitative detection of CPA was further achieved by a programming algorithm of gray value analysis, the corresponding regression equation, and a convenient smartphone app software (Figure 1). Our study establishes a fast, convenient, and visual detection method for foodborne pathogenic CPA, which provides a new idea for immunoassay of other soluble toxins.

## 2. Materials and Methods

### 2.1. Bacterial Strains and Plasmids

*Clostridium perfringens* type A standard strain (C57-1) was purchased from the China Institute of veterinary drug control. The constructed prokaryotic expression vector (PET30a) with the full gene of *C. perfringens* α toxin (GenBank accession: DQ202275) and α toxin 4 amino acid mutation (D56G, D130G, Y275N, D336N) plasmid (CPA_m4_) and the triplet repeat concatenated C-terminus of α toxin CPA_C3_), (247-370aa) were constructed by Xiaoyun Chen from the China Institute of Veterinary Drug Control.

### 2.2. Animals and Preparation of Rabbit Polyclonal Antibody

Two Specific Pathogen Free (SPF) adult rabbits were purchased from the Institute of Radiation Medicine of the Chinese Academy of Medical Sciences in Tianjin, and rabbits were maintained under specific pathogen-free conditions and handled following the guidelines for animal experiments of Tianjin University and the Chinese Academy of Medical Sciences. The purified protein was injected into the adult SPF rabbit, subcutaneously, at a concentration of 1 mg/mL each time. Moreover, rabbits were injected on weeks 1, 2, 3, 4. Blood was collected 5 days after the last immunization and the sera were separated to prepare the purified antibody by the method of precipitation with saturated ammonium sulfate (pH 7.0).

### 2.3. Construction of Clostridium Alpha Toxin N-Terminal Prokaryotic Expression Vector

The N-terminal truncated expression plasmid (PET30a-CPA_N_) containing the D56G, D130G mutation was constructed by using the 4 amino acid mutation plasmid (CPA_m4_) as a template, and the primers with the restriction enzyme site of BamH I and Xho I of 5′-GGATCCCATATGTGGGATGGCAAG-3′ and 5′-CTCGAGGGATGGGTCATTACCCTCGC-3′. The constructed plasmid was further verified by PCR detection and sequenced by Jin Weizhi Co. in China.

### 2.4. Purification of Recombinant Protein

The expression plasmids (CPA_C3_ and CPA_N_) were transferred into *Escherichia coli* BL21(DE3) competent cells and the single colony on the plates were picked for culturing in LB medium containing 100 U/mL ampicillin until the bacterial concentration was OD_600_ = 0.8, and added 1 mmol/L Isopropyl β-D-Thiogalactoside (IPTG) to induce protein expression at 16 °C for 8 h. The recombinant inclusion protein in the bacterial precipitated lysate was denatured with 8M urea, and purified with a nickel column; performed as previously described [23]. Finally, the purified protein was observed and analyzed by 12% SDS-PAGE (Appendix A).

### 2.5. Determination of Antibody Concentration and Antibody Titer by Bradford Method

The series concentration of 0.10, 0.08, 0.06, 0.04, 0.02 mg/mL bovine serum albumin (BSA) was prepared to develop the standard curve using Total Protein Quantitative Colorimetric Assay Kit (lot: E-BC-K168-S) for detecting the protein contents as manufacturer’s instruction (Elabscience Biotechnology Co. Ltd., Wuhan, China). The OD_590_ nm was measured and the standard curve between absorbance and BSA concentration was calculated for determining antibody concentration. The OD value of 10-fold series dilution (1:10, 1:100, 1:1000, 1:10000) of antibodies against recombinant toxins were measured and concentration was calculated according to the standard curve. Three biological replicates were made for each experiment.

### 2.6. Western-Blot Analysis

First, the previously purified recombinant protein was subjected to 12% SDS-PAGE electrophoresis, and then transferred to a methanol-activated polyvinylidene fluoride (PVDF) membrane (Millipore, Burlington, MA, USA), and then the membrane was blocked in 5% skimmed milk purified rabbit serum antibody (1:500 dilution) at 4 °C, respectively. The goat anti-rabbit HRP-conjugate was further incubated for 1 h at room temperature. Finally, we used the Pierce Electro-Chemi-Luminescence (ECL) Western Blotting substrate (Thermo Fisher Scientific, 81 Wyman Street, Waltham, MA, USA) to complete the immunoassay.

### 2.7. Indirect ELISA Analysis

The 5 μg/mL purified CPA_C3_ or CPA_N_ protein diluted with carbonate buffer (pH 9.6) was added into a 96-well plate at 100 μL/well and stayed overnight at 4 °C; the plate was washed with 0.1% Tween-20 PBS buffer three three times, and continued to block it with 1% BSA-PBST buffer at 37 °C for 30 min. The series dilution toxin antibody or negative rabbit serum were added at 100 μL/well. The HRP-labeled goat anti-rabbit IgG conjugate was added at 100 μL/well and incubated at 37 °C for 30 min. Finally, 100 μL O-phenylenediamine (OPD) substrate coloring solution (Beijing Solable Technology Co., Ltd., Beijing, China) was added to each well and incubated at 37 °C for 10 min, and then the stop solution was added to measure the value of OD_490_ with a microplate reader (Bio-Rad, Hercules, CA, USA). The antibody titer of rabbit anti-CPA_C3_, CPA_N_ was calculated by the formula: (Positive serum/Negative serum) P/N = (OD value of sample-OD value of blank slot)/(OD value of negative serum-OD value of blank slot). The maximum serum dilution ratio is defined as its ELISA titer as the P/N value is 2.1.

### 2.8. Preparation of C. perfringens Alpha Toxin and Quantification of CPA Enzyme Activity

The purified CPA was prepared as previously described [24]. A total of 40 mL of the α toxin supernatant from the culture medium of *C. perfringens* type A strain C57-1 in the anaerobic beef liver broth was precipitated with 60% ammonium sulfate, and dissolved in 1 mL of pH 7.5 Tris-HCl buffer, demineralized by dialysis, concentrated by concentration tube, and determined by SDS-PAGE. The lecithin hydrolysis test was used to replace the semi-lethal toxicity test [25]. The α toxin can hydrolyze lecithin into phosphorylated choline and 1,2-diacylglyceride, and form milky white turbidity after the reaction. The degree of turbidity is linearly related to the virulence of α toxin within a certain range. The fresh chicken egg yolk supplement, with a terminal concentration of 4% CaCl_2_, 18% Al(OH)_3_ gel, was mashed and placed in a 4 °C refrigerator for 12 h. The supernatant centrifuged at 720× *g* for 20 min was called lecithin. The lecithin was added into two-fold dilution α toxin respectively and incubated at 37 °C for 1 h. The treatment of bacterial medium added lecithin as a negative control. The absorbance (turbidity) at OD_550_ was detected, and compared with the same dose of the control group; 10 LD_50_ doses of α toxin can be reflected in the sixth dilution of 2-fold serial dilution, in which the hydrolysis reaction can still appear turbid. It can be calculated that each hydrolysis dilution is equivalent to 10 LD_50_/2^6^, which is LD_50_/6.4 toxin toxicity. The virulence of the bacterial solution is calculated according to the toxicity of each dilution equivalent to LD_50_/6.4 when the hydrolysis reaction occurs. If the number of dilution tubes that can react is x, its virulence is 2xLD_50_/6.4. Moreover, the test further shows that the lecithin hydrolysis reaction activity has a stable correlation with the virulence of the toxin, and the lecithin prepared in different batches does not direct effect the test results. The standard of virulence dose produced by *C. perfringens* should be 10 LD_50_ (mice weight 20~22 g) as described [25].

### 2.9. FITC-Labeled Antibody against C. perfringens Alpha Toxin

The FITC-labeled anti-CPA_N_ antibody was prepared by a semi-permeable membrane labeling method. The 10 mg/mL anti-CPA_N_ antibody and 1/20 amount of antibody FITC (Solable Technology Co., Ltd., Beijing, China) dissolved in carbonate buffer (0.025 M, pH 9.0) were mixed, and then put into a dialysis bag, and placed at 4 °C for 16–18 h in a carbonate buffer. Finally, the FITC-labeled anti-CPA_N_ antibody was on dialysis in 0.01 M, pH 7.2 PBS buffer for 4 h, and stored at −20 °C for later use.

### 2.10. Coupling of Carboxyl-Modified Silica Microspheres with Antibody against CPA_C3_

Moderate silica microspheres (diameter 10 μm, Tianjin Junyijia Technology Co., Ltd., Tianjin, China), add 33% EDC and 50% NHS, were mixed well, shaken at room temperature for 20 min, then washed with PBS, and centrifuged to remove the washing solution, for three times. The optimized conditions for antibody and microsphere coupling were then determined.

### 2.11. Optimization of Detection Conditions

The optimal coating concentration of CPA_C3_ antibody to microspheres (under the fixed capture concentration of 10 LD_50_ CPA) were determined by variable incubation time, temperature, and photograph fluorescence imaging through observation of fluorescence microscope, and continued to get the specific value through the gray value analysis. Similarly, the optimal blocking conditions were determined by variable incubation time and temperature. The dilution concentration (1:50, 1:100, 1:200 dilution of FITC-anti-CPA_N_ antibody conjugate) and the incubation time of FITC-labeled CPA_N_ antibody conjugate were also determined under different blocking solutions (5% skim milk-PBST-20, 5% BSA PBST-20 buffer) conditions. Triplicates were performed under different conditions.

### 2.12. Statistical Analysis

We used ImageJ (Version: 1.52a) software to convert the fluorescent image into an 8-bit grayscale image, and then reversed it, analyzed the parameters of the fluorescence area, threshold, grayscale maximum, minimum, and average value, and finally got the average grayscale value we needed. Finally, we calculated the regression equation between CPA concentration and fluorescence/grayscale value. Prism 8.0 (GraphPad Software Inc., La Jolla, CA, USA) software was used for all statistical analysis and graphing in the article.

For the production of the smartphone app, the algorithm that analyzed the gray value of the fluorescence image was compiled through the self-programming algorithm, and then the regression equation, according to the gray value and the toxin concentration was programmed into the app. Finally, successfully produced an app for detecting CPA. This APP is suitable for all android systems.

## 3. Results

### 3.1. Construction of Clostridium Alpha Toxin (CPA) Domain Expression Vector and Preparation of Polyclonal Antibody

The three-dimensional (3D) structure of *C. perfringens* α toxin was analyzed to design the N-terminal (1–249 aa) or C-terminal (247–370 aa) domain protein expression vector, respectively. The *C. perfringens* α toxins (CPA) include two functional domains, the α-helix-based N-terminal domain, and β-sheet-based C-terminal domain [26] (Figure 2). The N-terminus domain performs phospholipase C activity, where histidine residues are required for its phospholipase C activity. The C-terminal contains the 4 α helix structure and hydrophilic side chains, which are very important for the sphingomyelinase activity of α toxins [27]. N-terminal and C-terminal alone have no hemolytic activity and lethal activity, and only their synergy can have the activity of α toxin itself. At the same time, there are two active centers of α toxin, but there are some overlaps, and the two active sites are interactive [26].

Recombinant expression vector with 3 tandem repeats at C-terminal of CPA, named PET30a-CPA_C3_, and four amino acid mutations of CPA (D56G, D130G, Y275N, D336N) named PET30a-CPA_m4_, were constructed. The site-directed mutant toxin can greatly reduce the toxicity and lethality of CPA, so it is convenient for the subsequent preparation of rabbit serum antibodies without the lethality of the rabbit. After obtaining the recombinant protein of the N-terminal (1–249 aa) and C-terminal (247–370 aa) domain, we prepared their polyclonal serum antibodies.

The purified recombinant CPA domain protein (1 mg/mL) inoculated adult rabbits by subcutaneous injection for four times, with a one-week interval between the adjacent two. Four days after the last immunization, the blood was collected and the obtained serum was purified by saturated ammonium sulfate precipitation. The standard curve was established using the Bradford method (Figure 3a). The measured antibody concentrations were: PET30a-CPA_C3_ = 7.02 mg/mL; PET30a-CPA_N_ = 7.47 mg/mL. The purified antibody was further analyzed by Western-Blot and indirect ELISA method (Figure 3b–e). The ELISA test titers of the two serum antibodies were CPA_C3_ = 1:3200 and CPA_N_ = 1:6400.

### 3.2. Quantification of Clostridium Alpha Toxin Concentration

CPA was purified by 60% ammonium sulfate and determined by SDS-PAGE analysis to obtain the concentration of CPA (ng/mL) (Figure 4a). The CPA has two toxicities of lecithinase C and sphingomyelinase [26], which can hydrolyze lecithin. Referring to the Kaerber method of the animal alternative test of toxicity of CPA [25], each dilution is equivalent to LD_50_/6.4 toxin virulence (10 LD_50_/2^6^). After performing different batches of CPA lecithin hydrolysis experiments, the CPA toxic lethal dose was 10LD_50_. The CPA can hydrolyze lecithin into phosphorylated choline and 1,2-diacylglyceride, and form milky white turbidity after the reaction. The degree of turbidity is linearly related to the virulence of CPA within a certain range [24]. Here, the purified CPA was quantified by the toxicity determination alternative test. The optical density 590(OD_590_) of the diluted CPA protein mixing with lecithin was detected, and the lethal dose 50(LD_50_) of CPA was calculated by a curve equation between OD_590_ and CPA concentration (Figure 4b), so the concentration of CPA was further quantified to determine its corresponding relationship with the LD_50_. In our results, the native CPA concentration of 0.525 μg/mL corresponds to 10LD_50_. To facilitate writing and reading, the subsequent introduction uses LD_50_ to describe the concentration of CPA.

### 3.3. Fluorescent Labeling of CPA_N_ Antibody and Optimization of Silica Microsphere Detection Conditions

To develop a visual double-antibody sandwich fluorescent method for the detection of CPA, we needed the antibody linked to nanomaterial silica microspheres and another fluorescence-labeled antibody. Here, the carboxyl groups modified nanomaterial silica microspheres were selected for coupling to an antibody against recombinant PET30a-CPA_C3_ protein. The microspheres connected with the antibody were indirectly bound, and finally, another antibody labeled with fluorescence was added for binding. As a result, the antibody with the fluorescent label was indirectly connected to the microsphere, and the fluorescence intensity detected by the fluorescence microscope was the quantitative CPA concentration (Figure 5a).

The optimal coating concentration of CPA_C3_ antibody to microspheres under the fixed capture concentration of 10 LD_50_ CPA were determined by variable incubation time, temperature, and then photographed fluorescence imaging through observation of fluorescence microscope, and continued to get the specific value through the gray value analysis (Appendix A). Finally, the optimal coating antibody of 0.1 mg/mL, mixed with carboxyl-modified silica microspheres at 25 °C for 2 h, was determined. Similarly, the fluorescent image-gray value analysis of the optimal blocking conditions (Appendix A) and FITC-labeled CPA_N_ antibody conjugate concentration (Appendix A) were determined as 5% BSA for 12 h at 4 °C, 1:50 for 1 h at 37 °C, respectively.

To detect the fluorescent picture, we use ImageJ software to analyze the gray value, which is the fluorescence intensity of the figure (Figure 5b). Finally, according to the above experimental data, the suitable conditions for the fluorescence detection experiment were determined (Table 1). After interacting with the activated silica microspheres, with 0.1 mg/mL capture antibody (CPA_C3_) at 25 °C for 2 h, it was blocked with 5% BSA overnight at 4 °C. The next day, at 37 °C, the α toxin and the 0.2 mg/mL detection antibody (FITC-CPA_N_) interacted with the microspheres coupled with the capture antibody for 1 h, and the detection could be performed by fluorescence microscope.

### 3.4. Establishment of Clostridium Toxin Fluorescence Detection System and Production and Application of Smartphone APP

After determining the best experimental conditions, we conducted a formal fluorescent detection experiment of CPA. First, we needed to perform a gradient dilution of the concentration of CPA. The preliminary quantification in the previous (Result 3.2) determined that the concentration of undiluted protoxin was 10 LD_50_, so we successively diluted it two times to obtain 10LD_50_, 5LD_50_, 2.5LD_50_, 1.25LD_50_, 0.625LD_50_, and 0.3125 LD_50_. Three biological replicates were performed for each concentration gradient. Through ImageJ analysis of the gray value, it was found that the fluorescence intensity of 0.3125LD_50_ dilution was too weak, and the analyzed gray value was too low to form a linear relationship (average gray value = 0.003). The corresponding fluorescence picture obtained by fluorescence microscopy (Figure 6a, and Appendix A). The ImageJ software was used to analyze the gray value, and finally draw the regression curve between the obtained gray value and different toxin content (Figure 6b, Appendix A), the regression equation: Y = 0.01915X + 0.002875 (R^2^ = 0.9972). This method was used to determine the minimum detection limit of CPA, 0.625 LD_50_, with high sensitivity, and the standard deviation of the gray value of the lowest detection limit of 0.625LD_50_ was 0.0017. Afterward, the five dilutions of the detected toxin were divided into five groups, and three biological replicates were carried out for each dilution. The gray value was analyzed for the coefficient of variation (Appendix A, Table 2). The analysis showed that the results were all low at 15%, which indicated that the same sample had a low degree of variation in the same batch and the established fluorescence microsphere detection CPA method had good stability, and provided a reliable premise for the subsequent programming of the smartphone app.

Finally, a combination of regression equation and image processing technology was used to make a mobile app through programming (Figure 7). This software program is suitable for android smartphones that include the basic parameters settings: image name, average gray value, and value of LD_50_. The specific operation was to upload the fluorescent picture we detected to the mobile phone app, and obtain the corresponding CPA concentration in real-time, immediately.

*Clostridium perfringens* causes an acute infectious disease in livestock by ingesting of the drinking water or feed contaminated by *Clostridium perfringens*. Besides, *Clostridium perfringens* grows rapidly in milk, which can lead to milk pollution. Therefore, after the successful production of the smartphone app, we used CPA-negative milk, and artificially simulated CPA milk was used to evaluate the effectiveness of the established method in this experiment according to the recommendation of GB 4789.13-2012. The different concentrations of CPA-contaminated milk (0.625LD_50_, 2.5LD_50_, 10LD_50_) were prepared, and then the mixture of the centrifugation supernatant of the milk CPA_C3_-antibody coating microspheres, CPA_N_ antibody-FITC conjugate interacted for 20 min, and then performed the microsphere fluorescence detection experiment that we introduced above. The fluorescence image was transferred to the app to analyze the image fluorescence intensity and the CPA concentration. The recovery rate, ranging from 80% to 108.8%, was obtained according to the standard regression equation (Table 3).

## 4. Discussion

*C. perfringens* can be classified into seven types, A, B, C, D, E, F, and G, according to the main lethal toxins α, β, ε, ι, CPE, and NetB [28]. Among them, α toxin of *C. perfringens* (CPA) is universally present in all strains and is a kind of multifunctional metal enzyme that depends on zinc ion. It has two enzyme activities of phospholipase C and sphingomyelinase and can hydrolyze phosphatidylcholine and sphingomyelin simultaneously [29]. CPA relies on these two enzyme activities to hydrolyze the membrane phospholipid that is the main component of the cell membrane, thereby destroying the integrity of the cell membrane structure and causing cell lysis. It has the characteristics of cytotoxicity, hemolytic activity, skin necrosis, platelet aggregation, and increased vascular permeability [3]. In summary, this experiment uses the ubiquitous α toxin to establish a detection system for *C. perfringens,* and uses a representative standard strain of *C. perfringens* type A (C57-1 strain) as the research object of the entire experiment.

At the same time, according to related reports, in mature α toxins, aspartic acid at positions 56 and 130 are Zn^2+^ binding sites, and mutations in the above two amino acid sites can significantly reduce the activity of α toxins [30]. Alberto et al. found that the hemolytic activity of Y275N and D336N mutants reduced to 11% of the wild type toxin and the cytotoxicity was reduced to 1/100,000 and 1/1000 of the wild type toxin, respectively [31]. Many reports were done on the preparation of *C. perfringens* vaccines by introducing single amino acid mutations to reduce the potential toxicity in large-scale production. In our study, the amino acid sites of *C. perfringens* type A (C57-1 strain) CPA protein 56, 130, 275, and 336 aa were mutated to prepare the recombinant avirulent protein, which is beneficial to animal immunity and preparation of the antibody. The results also confirmed that the reduced toxicity of CPA and both of CPA_m4_ and CPA_N_ keep good immunogenicity for preparing polyclonal antibodies. The antibodies can effectively be binding native CPA, for developing a double antibody sandwich microsphere method to detect CPA.

At present, the detection of *C. perfringens* is mainly based on molecular biology inspection PCR or the ELISA method [32,33]. Olsvik et al. [34] established a 4-layer sandwich method to detect 0.1 μg/mL *C. perfringens* enterotoxin for four days to complete the whole process. The indirect ELISA method established by McClane can detect 25 ng/mL enterotoxin and apparently shorten in the experimental period [35]. An antigen-capture ELISA method can detect 19 ng/mL *C. perfringens* α toxin (CPA) [13]. These methods can replace the serum neutralization test for the diagnosis of *C. perfringens*. Uzal et al. used PCR technology to detect the α toxin contents of goat feces and the gastrointestinal tract successfully, but the method is cumbersome and uneconomical [36]. Yoo established a multiplex PCR to detect four major toxins of *C. perfringens*, but this method requires the extraction and purification of bacterial genomes [37]. The fluorescence quantitative PCR established by Gurjarcan directly detect the major toxin genes of *C. perfringens* from fecal samples, and can be used for epidemiological investigation of *C. perfringens* infection [38]. In summary, the various detection technologies established at present have problems, such as time-consuming, cumbersome methods, uneconomical, or low sensitivity.

The basic principle of the detection system for *C. perfringens* established in this research is to use the unique bifunctional domain of α toxin to prepare specific SPA structure domain antibody, with the two antibodies of the N-terminal and the C-terminus of CPA to sandwich the α toxin in the sample. We label one of the antibodies (CPA_N_) with the fluorescent dye FITC so that it can be detected more intuitively and can be quantified based on the fluorescence intensity. Fluorescence-modified silica nanoparticles are widely used in hypersensitivity detection, drug-loading fields, due to their small size effect, surface effect, quantum size effect, and good dispersion performance. Silica nanomaterials have unique properties in other aspects, with their excellent stability, reinforcement, thickening, and thixotropy, and have unique characteristics in many disciplines and fields, and have an irreplaceable role [39,40]. Therefore, silica nanomaterials modified with specific recognition molecules have shown a wide range of application prospects of pathogen detection, nano-drug loading, tumor treatment, biomarkers, etc. In our study, we choose carboxyl modified silica microspheres as the carrier and coupled the antibody CPA_C3_ on it to realize the enrichment of CPA toxin in a food sample, using the FITC-labeled CPA_N_ to amplify the detection signal. Then, fluorescence microscope observation and fluorescence intensity analysis were performed by establishing a regression curve between the concentration of CPA and fluorescence images. The establishment of the detection system had good stability (variation coefficient <15%) and high sensitivity (lowest detection limit is 0.625 LD_50_, i.e., 32.8 ng/mL), and could use *C. perfringens* contamination evaluation based on the good water solubility of CPA. Moreover, the experiment period was relatively short. We incubated the captured antibody silica microspheres, centrifugation supernatant for processing food samples, and the fluorescence-labeled CPA_N_ for 60 min. Finally, by getting the fluorescence image by fluorescence microscope, subsequent data processing, and analysis was performed quickly. The total detection time was less than 90 min.

Moreover, we have further improved the visual operation based on the established detection technology. Through computer programming, the image processing technology and the constructed linear regression equation were combined to make the universal app software for android devices. We can get the CPA concentration of food samples directly by app recognition of the transferred fluorescence image obtained from the experiment. The biggest advantage of this visualization technique is that it can greatly shorten the processing time of experimental data. After getting the fluorescence picture, we can upload it directly to the app to get the results we need immediately. It is convenient, easy to operate, and only needs an android smartphone to get results anytime, anywhere.

In summary, the smartphone combined with the nano-microsphere fluorescence detection system for detection of CPA established in this research has good stability and high sensitivity, and the detection method is fast, simple, and portable, which can be broadened to the detection of other toxin produced by food-contaminated bacteria. The convenient and intelligent detection system will have good application prospects for various detection fields (food, environment, clinical, etc.) in the future.

## 5. Patents

The detection system for the alpha toxin of *Clostridium perfringens* established in this study has been patented (patent No:202010883292.3).

## Figures and Tables

**Figure 1 microorganisms-08-01865-f001:**
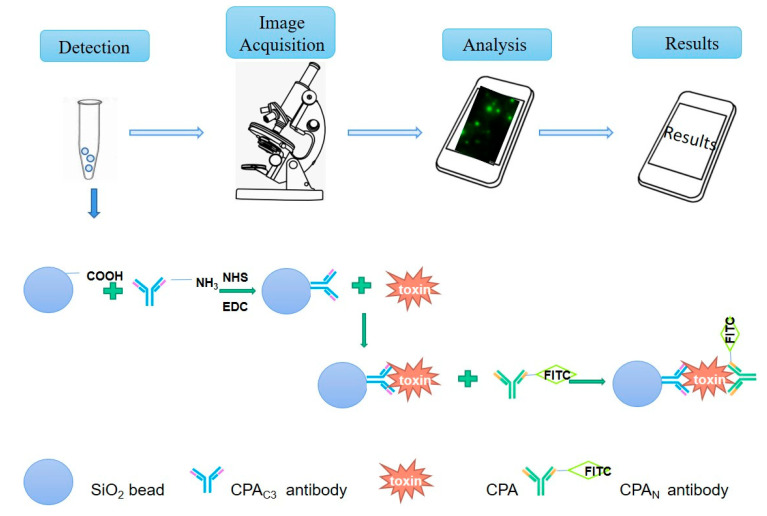
The principle of silica microsphere conjugated fluorescent antibody combined with smartphone app detection system.

**Figure 2 microorganisms-08-01865-f002:**
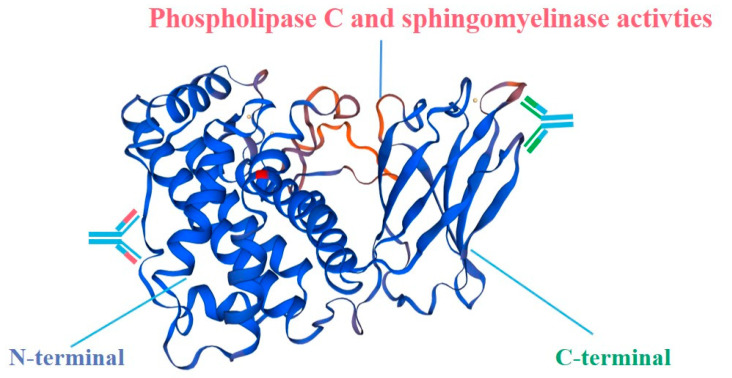
The three-dimensional (3D) structure of *C. perfringens* alpha toxin (CPA). Based on the 3D structure to design the N-terminal (1–249 aa) and C-terminal (247–370 aa) domain protein expression vector. The CPA includes two functional domains, the α-helix-based N-terminal domain and the β-sheet-based C-terminal domain [2]. The N-terminus has phospholipase C activity, where histidine residues are required for α toxin phospholipase C activity. The C-terminal contains the 4 α helix secondary structures and hydrophilic side chains, which are very important for the sphingomyelinase activity of α toxins [1]. N-terminal and C-terminal alone have no hemolytic activity and lethal activity, and only the synergy of the two can have the activity of α toxin itself. At the same time, there are two active centers of α toxin, but there are some overlaps, and the two active sites are interactive [2].

**Figure 3 microorganisms-08-01865-f003:**
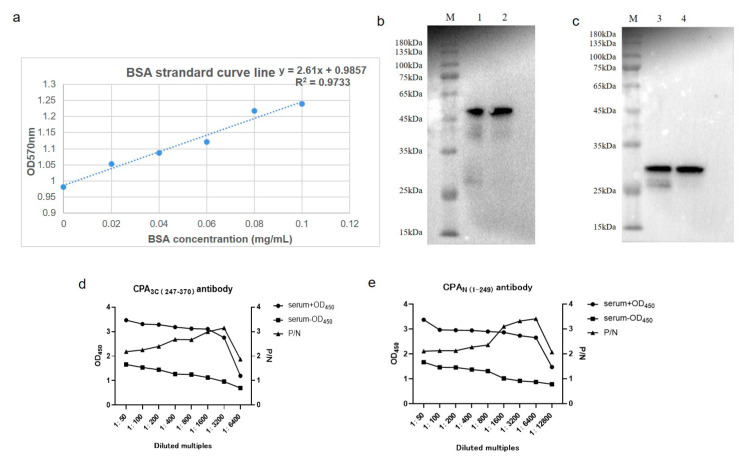
Determination of polyclonal antibody concentration and titer of *C. perfringens* alpha toxin (CPA_C3_ and CPA_N_). (**a**) Standard curve of antibody concentration measured by Bradford method; (**b**,**c**) Western-blot results of recombinant proteins CPA_C3_ and CPA_N_(M: protein marker (GenStar M221) lane 1–2: CPA_C3_ recombinant protein, lane 3–4: CPA_N_ recombinant protein). (**d**,**e**) Indirect ELISA results of recombinant proteins CPA_C3_ and CPA_N_: the titer of the detected CPA_C3_ antibody is 1:3200, and the titer of the detected CPA_N_ antibody is 1:6400. The antibody titer of rabbit anti-CPA_C3_, CPA_N_ was calculated by the formula: P/N = (OD value of sample—OD value of blank slot)/(OD value of negative serum- OD value of blank slot). The maximum serum dilution ratio is defined as its ELISA titer as the P/N value is 2.1.

**Figure 4 microorganisms-08-01865-f004:**
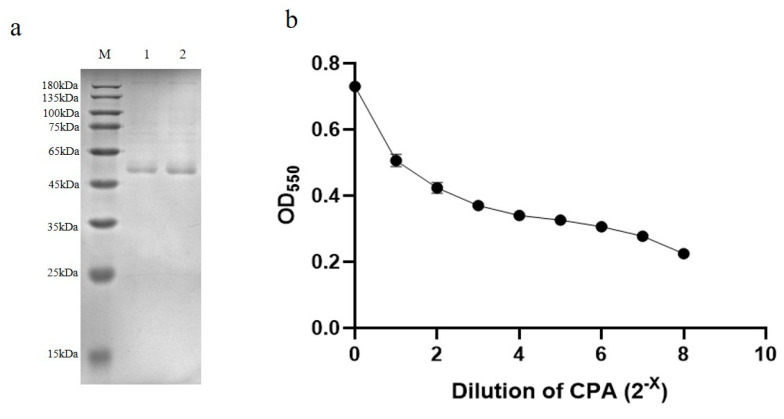
Crude purification and quantification of CPA (**a**) SDS-PAGE results of CPA crude purification: M: protein marker (GenStar M221), lane 1:0.21 µg/mL crude purified CPA (40 times concentrated), lane 2: 0.42 µg/mL crude purified CPA (80 times concentrated); (**b**) degree of hydrolysis of CPA by lecithin hydrolysis method.

**Figure 5 microorganisms-08-01865-f005:**
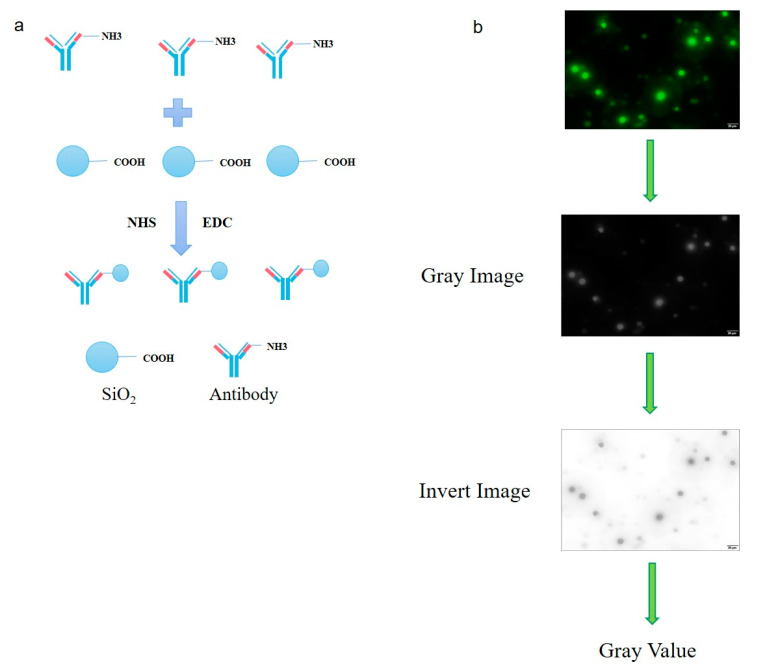
Coupling of silica microspheres and antibodies and gray value analysis of fluorescence Images. (**a**) Principle of coupling between silica microspheres and antibodies: under the catalysis of NHS (N-Hydroxysuccinimide) and EDC (1-(3-Dimethylaminopropyl)-3-ethylcarbodiimide hydro), selecting the carboxyl modified silica microspheres is covalently bound to the amino group of the serum antibody prepared by us. (**b**) Fluorescence intensity analysis of fluorescence images: Image J software used to analyze the gray value of fluorescence images. Scale bar = 50 μm.

**Figure 6 microorganisms-08-01865-f006:**
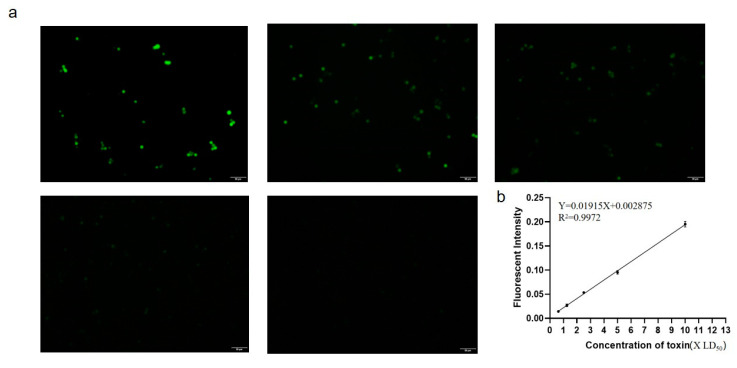
Fluorescence picture of CPA detection and the established standard regression equation. (**a**) Fluorescence pictures at different toxin concentrations: from left to right concentrations are 10LD_50_, 5LD_50_, 2.5LD_50_, 1.25LD_50_, 0.625LD_50_; (**b**) Standard regression equation based on fluorescence intensity and toxin concentration, Y = 0.01915X + 0.002875 (R^2^ = 0.9972). Scale bar = 50 μm.

**Figure 7 microorganisms-08-01865-f007:**
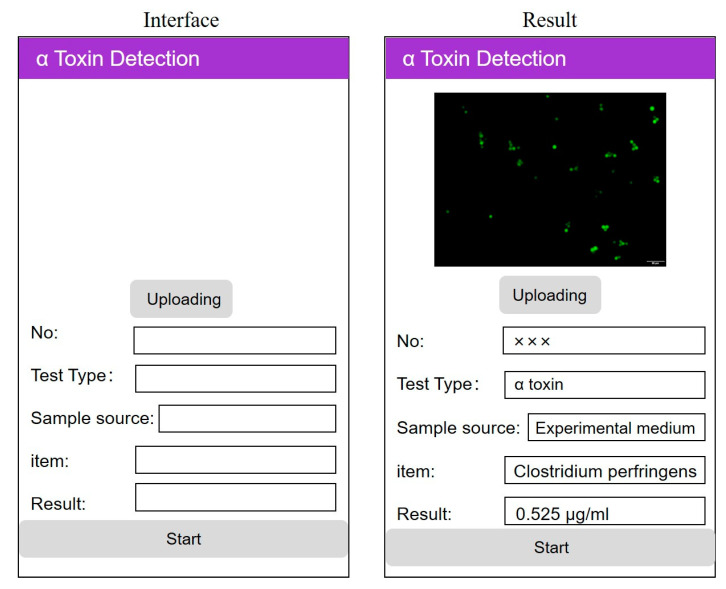
Smartphone APP detection interface. Scale bar = 50 μm.

**Table 1 microorganisms-08-01865-t001:** Best detection conditions for fluorescent microsphere detection experiments.

	Capture Antibody	Blocking Condition	Detection Antibody
Optimal dilution	0.1 mg/mL	5% BSA	0.2 mg/mL
Reaction conditions	25 °C, 2 h	4 °C, 12 h	37 °C, 1 h

**Table 2 microorganisms-08-01865-t002:** Repeatability test for fluorescence detection of silica microspheres (*n* = 3).

Sample Group (3/group)	Average MN ± SD	Coefficient of Variation CV (%)
1	0.195 ± 0.0048	2.46
2	0.095 ± 0.0098	10.3
3	0.053 ± 0.0018	3.40
4	0.027 ± 0.0022	8.14
5	0.014 ± 0.0017	12.1

**Table 3 microorganisms-08-01865-t003:** Recovery rates of different alpha toxin concentrations in milk samples.

Sample	Toxins	Add Concentration (×LD_50_)	Detected Concentration	Recovery (%)
1	CPA	0.625	0.50	80
2	CPA	2.5	2.72	108.8
3	CPA	10	10.24	102.4

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
