# Peer review of "Visual Detection of Clostridium perfringens Alpha Toxin by Combining Nanometer Microspheres with Smart Phones"

_microorganisms, 2020, doi:10.3390/microorganisms8121865_

Round 1
Reviewer 1 Report
The reference in the introduction is inappropriate, and some references cannot be searched, so please check it once. Please check the introduction again, and describe correctly. This paper is in the pre-review stage.
Author Response
Thank very much for your precious comments to our manuscript. All references are reorganized and formatted using MDPI's EndNote template. We have revised and updated our introduction section. The fuzzy description and grammatical errors have also been modified.
Reviewer 2 Report
The manuscript describes a new detection method for C. perfringens alpha toxin using nanospheres with smart phone app. This is an interesting topic and should be of interest to many readers interested in new detection technologies.
Comments:
- Needs extensive English usage (grammar and punctuation) editing. This is a must, as the manuscript is very hard to understand “as is”.
- Perhaps use ”Visual” instead of “Visualization?
- Line 18 and 19. Less than 90 min…sentence. Not sure what you mean? And the open parenthesis? Do you mean to say the new assay takes 90 min to perform and can detect as little as 0.625 LD50? Also, it’s best not to give concentration based on LD50 as most people do not know what the LD50 for alpha toxin is. It’s best to give a number in ng/mL.
- Line 82. What is a bacterial antibody?
- Line 108 and line 222. Not sure what you mean by “a week interval between each adjacent two times”? it’s 4 injections. You can say rabbits were injected on weeks 0, X, Y, Z.
- Line 165-166 and lines 233-235 This line is hard to understand “10 LD50 dose of toxin hydrolysis dilution is 6, each dilution is equivalent to LD50/6.4 toxin toxicity” What is the unit for 6 and 6.4. I understand it’s lecithin activity. The lines 233-235, how does the 2x serial dilution give the same 6 and 6.4?
- Lecithin activity vs mouse bioassay. How consistent is this assay compared to the actual animal bioassay? And do they compare across laboratories?
- Line 231. Not sure what you mean “21μg/ml purified native CPA was gotten by SDS-PAGE results”. Did you mean to say that the concentration of CPA was determined by SDS-PAGE analysis?
- Line 233. Capture concentration of 10 LD50. Would different batches of toxin be different? what if the toxin is inactivated and would have more of the inactivated toxin around at the same LD50 which could absorb more of the antibodies?
- Line 275-276. Is this a linear correlation of lecithin activity and LD50? Usually with animals it is not linear.
- Line 293. Why milk? Is this the matrix where CPA is found?
- Figure 3. Define P/N here so does not need to look for it on the m&M
- Figure 5. How do you choose the frame to measure? Are areas selected randomly? Are they evenly distributed?
- Line 366 “only 11% of wild type toxin” not sure what you mean and what you are referring to?
- Lines 407, 408 “lowest detection limit is 0.625 LD50, i.e. 32.8 ng/mL”. So what is the standard error/deviation?
Reviewer 3 Report
The manuscript describes a method, using an APP software for Android devices, to detect and quantify alpha-toxin from Clostridium perfringens (CPA). The method requires a fluorescent microscope to capture nanoparticle images (after the incubation protocol) that have to be further processed with ImageJ software before the analysis with the mobile APP. It is assumed that the detection of CPA has been used as a model to run and show the application, but it can be used for the detection and quantification of other molecules of interest, whether adequate antibodies are provided. It represents an elegant experimental approach, with accurate and precise methodology, that could be considered as a technical report and suitable for a more specialized journal.
Comments.
CPA ids produced by all the C. perfringens types, thus it is not specific to define a particular toxinotype. One of the limitations of the method is that it detects and measures only the presence of alfa-toxin, not the C. perfringens toxinotype. Accordingly, the present method should not be compared with multiple detection methods of C. perfringens toxins or toxinotypes.
As stated, before analyzed with the mobile APP, images should be obtained and processed with a fluorescent microscope and ImageJ software respectively. Using this equipment, a regression equation is obtained to be used by the mobile APP to calculate the CPA concentration. Would not be simpler and faster to use the ImageJ software to directly calculate the sample CPA concentration? Which is the real advantage to use the APP instead?
Additional comments.
- There are significant differences comparing diagrams of figure 1 and figure 5a, showing the silica microspheres and antibodies coupling approach (note the part of antibody bound to the microsphere). Also, to which extent figure 5b can be compared to the diagram shown in figure 5a? Are the microspheres fluorescent? It would probably be more suitable to add the images of figure 5b to the general scheme of figure 1 (and delete figure 5a).
- How the protein marker GenStar M221 was visualized in the western-blot images of figure 1?
- Line 352. The classification of Clostridium perfringens toxinotypes has been recently updated (Rood et al. Anaerobe 53: 5-10, 2018, doi:10.1016/j.anaerobe.2018.04.011 ).
- Supplementary tables should be revised:
Table S3. “fluorescent-labeled antibody” instead of “enzyme-labeled antibody”; what 150, 1100 and 1200 in the “Average gray value” mean?
Table S4 How the mean was calculated? Was from the “Gray value (triplicate)”?
Line 258, should be supplement table 2 (instead of supplement table 3)
Line 259, should be supplement table 3 (instead of supplement table 4)
- Figure A2. A histogram comparing the two blocking solutions (skimmed milk or BSA) would be more appropriated.
- Authors claim the total detection time (full methodology?) can last less than 90 minutes (possibly plus 30 minutes for coupling and incubations), but the general protocol shown in lines 261-269, which represents the optimal conditions (Table 1), is much longer. Which step (or steps) should be shortened to have similar good results (resolution, sensitivity, specificity) of the depicted optimal method?
- Please, unify the terminology for milliliter (for instance Ml in table 1 and ml in lines 231, 266, 326, 327, 376, 378, 379)
- English should be revised, mainly in the Material and Methods section
Round 2
Reviewer 1 Report
In the manuscript (microorganisms-990502), Cao et al. have developed a highly sensitive detector for alpha-toxin from Clostridium perfringens. I think that this work has been well done and convincing and the results shown in this paper have been also clearly demonstrated. The manuscript is also revised accurately. I judged therefore that this paper is suitable for publication of Microorganisms.
Author Response
Response 1: Thank very much for your precious comments and valuable suggestions to our manuscript. The English grammatical errors, phrases have also been polished.

Reviewer 2 Report
The answer to comments and edits are acceptable.
Author Response
Response 1: Thanks for your comments and valuable suggestions. The English grammatical mistakes, words and phrases have been corrected and polished in our update manuscript.
Reviewer 3 Report
Most of the concerns have been solved in the new version of the manuscript.
Just two additional comments:
Why name Clostridium perfringens as Clostridium welchii in lines 336-338? (Yes, Clostridium perfringens can be named Clostridium welchii, but why just in these lines?, why not to continue naming it as Clostridum perfringens as in the rest of the manuscript?
English has been improved in some parts of the manuscript, but it requires additional review.
Author Response
Most of the concerns have been solved in the new version of the manuscript. Just two additional comments:
Why name Clostridium perfringens as Clostridium welchii in lines 336-338? (Yes, Clostridium perfringens can be named Clostridium welchii, but why just in these lines?, why not to continue naming it as Clostridum perfringens as in the rest of the manuscript?
Response 1: Thank you for your comments. According to your suggestions, we have revised the sentence and changed Clostridium welchii as Clostridium perfringens.
English has been improved in some parts of the manuscript, but it requires additional review.
Response 2: The English grammatical mistakes, words and phrases have been corrected and polished in our update manuscript.
